# Nutritional Challenges in Older Cancer Patients: A Narrative Review of Assessment Tools and Management Strategies

**DOI:** 10.3390/nu17182928

**Published:** 2025-09-11

**Authors:** Giulia Giordano, Roberta Terranova, Luca Mastrantoni, Francesco Landi

**Affiliations:** 1Department of Geriatrics, Orthopedics and Rheumatological Sciences, Fondazione Policlinico Universitario IRRCS A. Gemelli, 00168 Rome, Italy; francesco.landi@unicatt.it; 2Geriatrics, Università Cattolica del Sacro Cuore, 00168 Rome, Italy; rob.terranova@libero.it; 3Medical Oncology, Università Cattolica del Sacro Cuore, 00168 Rome, Italy; luca.mastrantoni01@icatt.it

**Keywords:** malnutrition, geriatric oncology, elderly

## Abstract

**Background/Objectives**: Malnutrition, sarcopenia, cachexia, and frailty often coexist in older cancer patients and are associated with worse treatment tolerance, reduced quality of life, and increased mortality. These syndromes can be underrecognized, and the therapeutic approach is often fragmented. In light of this, the aim of this review was to synthesize current evidence on the screening and clinical management of nutritional aspects and the related tools, favoring multidimensional and personalized nutritional care. **Methods**: This narrative review was conducted according to the SANRA guidelines. A comprehensive literature search was performed on PubMed for studies published between January 2000 and June 2025, with no language restrictions. Eligible studies included adults aged ≥65 with cancer, addressing malnutrition, sarcopenia, cachexia, frailty, or nutrition-related interventions. **Results**: Malnutrition affects 30–80% of older cancer patients and is strongly associated with reduced survival, impaired treatment tolerance, and poorer quality of life. Tools such as PG-SGA, G8, GNRI, and CONUT offer practical options for early risk identification. Nutritional interventions, including oral supplements, dietary counseling, symptom management, and multimodal strategies (nutrition plus exercise), are associated with improved clinical outcomes. Evidence also supports the prognostic value of early screening and individualized nutrition care pathways. **Conclusions**: Malnutrition represents a modifiable risk factor in geriatric oncology and should be assessed considering other related conditions such as sarcopenia, cachexia, and frailty. Systematic screening and targeted interventions should be integrated into standard cancer care to improve outcomes in older adults. Future research should prioritize personalized nutrition strategies and multicenter trials focused on survival, function, and quality of life.

## 1. Introduction

Cancer is the second leading cause of death worldwide, contributing significantly to both morbidity and mortality, particularly in the older population [1]. Cancer and aging are interconnected processes, as most malignancies occur in older adults, who often have reduced physiological reserves and a higher burden of comorbidities [2]. Within this context, malnutrition emerges as a multifactorial major geriatric syndrome and a prognostic factor in oncology. It can result from inadequate dietary intake, catabolic effects of tumors, age-related anorexia, and social or cognitive factors [3]. The risk of malnutrition itself in an older person can be related to multiple negative outcomes, such as functional decline, higher toxicity from chemotherapy, poorer response to treatment, and increased mortality [4]. Still, nutritional assessment and intervention are often underprioritized amidst the complexities of oncological treatments [4]. Moreover, older adults can also be affected by other conditions such as cachexia, sarcopenia, and frailty, potentially magnifying malnutrition and subsequently affecting patient’s outcomes in terms of quality of life, treatment tolerance, adverse events, and survival. These syndromes tend to be underrecognized, and the therapeutic approach is often fragmented, as there is a lack of standardization of the screening phase and of integrated, multi-modal clinical approaches.

In this context, the aim of this narrative review is to synthesize the prevalence and clinical impact of malnutrition and related conditions such as sarcopenia, cachexia, and frailty in older adults with cancer, to summarize the available tools for nutritional screening and assessment, and to review the evidence on clinical management and interventions, thus supporting clinicians in the implementation of early screening and favoring multidimensional and personalized nutritional care.

## 2. Materials and Methods

This narrative review was conducted according to the SANRA (Scale for the Assessment of Narrative Review Articles) guidelines (Appendix A) [5].

### 2.1. Search Strategy and Data Source

A comprehensive literature search was performed on PubMed for studies published between 1 January 2000 and 30 June 2025, with no language restrictions, focused on articles involving human subjects, with attention to populations aged ≥65 years. The search strategy included the following terms: (“malnutrition” OR “nutritional status” OR “nutritional assessment” OR “cachexia” OR “sarcopenia” OR “frailty”) AND (“older adults” OR “elderly” OR “geriatric patients”) AND (“cancer” OR “neoplasm” OR “oncology”) AND (“nutrition intervention” OR “Mediterranean diet”). Additional sources were identified through manual searches of reference lists from key reviews and meta-analyses. The literature search was conducted by two authors (GG and RT), and studies were included by consensus. Disagreements were resolved referring to another author (LM). The initial search yielded 523 articles, which were screened for relevance based on titles and abstracts. Relevant studies were assessed by full text for inclusion.

### 2.2. Inclusion and Exclusion Criteria

The inclusion criteria were as follows: (1) studies involving adults ≥ 65 years with cancer, (2) focus on nutritional status, screening tools, and syndromes (cachexia, sarcopenia, frailty), or nutrition-related interventions, and (3) RCTs, observational studies, reviews, or guidelines (ESPEN, ESMO, SIOG, ASCO). The exclusion criteria included (1) preclinical or animal studies and (2) articles with insufficient methodological clarity or outcome data.

### 2.3. Data Extraction and Synthesis of Results

Data extraction was performed by two authors (GG and RT); disagreements were resolved by consensus or referring to another author (LM). For each study, information such as study design, population characteristics, cancer type, nutritional assessment methods, and interventions (when applicable) was considered. The studies were narratively synthesized, and given the narrative nature of this review, no formal meta-analysis nor risk-of-bias assessment was conducted.

## 3. Malnutrition, Sarcopenia, Cachexia, and Frailty: Definitions and Overlap

### 3.1. Malnutrition

According to the 2019 Global Leadership Initiative on Malnutrition (GLIM), malnutrition is defined by the presence of phenotypic criteria (weight loss, low body mass index, or reduced muscle mass) and etiologic criteria (reduced food intake/assimilation or disease burden/inflammation) [6]. Many cancer patients meet these criteria due to tumor-induced hypermetabolism and treatment side effects, such as mucositis or nausea, that impair nutritional intake. In older cancer patients, malnutrition could manifest as involuntary weight loss, muscle wasting, and reduced dietary intake, accompanied by an inflammation status.

### 3.2. Cachexia

A closely related concept is cancer cachexia, a syndrome consistent of weight loss and muscle mass loss, driven by the tumor and systemic inflammation [7]. Cachexia is defined as >5% weight loss (or BMI < 20 with >2% weight loss) in the context of chronic illnesses such as cancer, often accompanied by anorexia and elevated inflammatory markers [8]. It is estimated that 50% of all cancer patients develop cachexia, with the highest incidence (up to 80%) in gastrointestinal malignancies such as gastric and pancreatic cancers [9]. The prevalence of cachexia in older adults is between 52% and 62%, and sarcopenia could also be present in approximately 57% [10].

Cachexia represents a hypercatabolic condition that cannot be fully reversed by conventional nutritional support alone, and it can contribute directly to mortality, with an estimated 20–30% of cancer deaths attributable to this specific wasting syndrome. Dunne et al. found that cachexia was associated with impairment in instrumental activities of daily living (IADLs), while sarcopenia and isolated weight loss were not significantly linked to IADL impairment [11]. Furthermore, cachexia was significantly associated with poorer survival and reduced tolerance to chemotherapy (requiring dose reductions), with an increased risk of experiencing more severe side effects [12].

### 3.3. Sarcopenia

Sarcopenia is the age-related loss of skeletal muscle mass and strength, which can be accelerated by malignancy and its treatments. It is characterized by low muscle mass (assessed by imaging or other methods such as bioimpedance) plus evidence of low muscle function, evaluated through grip strength, the chair stand test, or gait speed, in accordance with the EWGSOP2 consensus criteria [13]. While this approach generally relies on pre-specified population-based cut-offs, efforts towards personalized predictions were recently developed, aiming to predict reference quantiles through machine learning models [14].

In older cancer patients, sarcopenia and cachexia could overlap considerably, since both involve muscle wasting. However, sarcopenia can also occur in patients who are weight-stable or obese—a condition known as sarcopenic obesity—and is not necessarily driven by inflammation, which distinguishes it from cachexia [8]. The reported prevalence of sarcopenia among cancer patients varies widely, depending on the assessment method, ranging from approximately 5% to 89% [8]. Referring to modern criteria, it is estimated that 20–50% of older adult with cancer are typically sarcopenic. In a cohort of >3000 colorectal cancer patients, 58% of those over 70 years had sarcopenia, compared to around 27% of patients under 60 [15].

### 3.4. Frailty

All the above conditions—malnutrition, cachexia, and sarcopenia—could reciprocally influence and amplify their effects, possibly leading to frailty. Frailty is a syndrome consisting of a state of increased vulnerability to several stressors and often coexists with malnutrition and sarcopenia in older patients with cancer. Frailty has been defined by Fried as meeting three out of five phenotypic criteria: low grip strength, low energy, slowed walking speed, low physical activity, and/or unintentional weight loss [16]. In older cancer patients, frailty prevalence is high: large surveys indicate only about 40% are “robust,” while the remaining 60% are either pre-frail or frail. For instance, an analysis of >3900 US cancer survivors aged ≥65 years found that 23.8% were frail, and another 35.9% were pre-frail [17]. Frailty is an independent predictor of worse outcomes, including higher rates of chemotherapy discontinuation, postoperative complications, longer hospitalizations, and mortality [17]. Frail patients with cancer often experience lower treatment tolerance and greater risk of treatment-related toxicity. Furthermore, frailty is linked to an increased risk of post-surgical complications, including infections, prolonged hospital stays, and delirium, and it has been identified as an independent predictor of mortality in older adults with cancer [18,19,20]. A frail patient is also at a greater risk of premature treatment discontinuation and reduced survival. 

A summary of key points regarding malnutrition, cachexia, sarcopenia, and frailty is available in Table 1.

## 4. Prevalence and Clinical Impact of Malnutrition

Malnutrition is highly prevalent in older people with cancer, with rates ranging from 30% to 80% across studies [4]. Specific subgroups, such as older patients undergoing chemotherapy and patients with gastro-intestinal tract tumors, seem more vulnerable: these patients could manifest altered gastrointestinal absorption leading to Vitamin D deficiency, osteoporosis, an increased risk of falls, and fractures [22].

Malnutrition emerged as an independent predictor of worse overall survival (OS) in numerous analyses. A retrospective study found malnourishment was a risk factor for all-cause mortality in older cancer patients, both in univariate (HR, 1.49; 95% CI, 1.08–2.05; *p* = 0.01) and multivariate analyses for older patients with solid tumors, where malnutrition increased the risk of all-cause mortality (HR, 1.87; 95% CI, 1.10–3.17; *p* = 0.02) [23] and was also related to longer hospital stay and higher rate of hospital admission

Malnutrition is also associated with functional impairment and poorer physical performance, both worsening QoL [24]. Nutritional impairment and lower physical function have also been correlated with higher symptom burden: a study found that a deficiency in the iADL was associated with higher risk for malnourishment [25], while other evidence reported that patients with functional impairment exhibit poorer nutritional status [26].

A cross-sectional study involving 336 older adults (mean age 70 ± 7.2 years) used a geriatric assessment (GA) along with the Cancer & Aging Resilience Evaluation (CARE) survey, which includes nutritional evaluation. The most common tumor types included in the study were colorectal (33.6%) and pancreatic cancer (24.4%). Nutrition scores were dichotomized into normal (0–5) and malnourished (≥6), and multivariate analyses, adjusting for demographics, cancer type, and cancer stage, were used to examine associations with GA impairment, health-related quality of life (HRQOL), and healthcare utilization. Overall, 52.1% of the patients were identified as malnourished. Malnutrition was related to a higher prevalence of several GA impairments, such as ≥1 falls with adjusted odds ratio (aOR) of 2.1, iADL impairment (aOR = 4.1), and frailty (aOR 8.2). Malnutrition was also associated with HRQOL domains, both physical (aOR = 8.7) and mental (aOR = 5.0), and with prior hospitalizations (aOR = 2.2) [27].

It is important to note the overlapping nature of these conditions: many older cancer patients show concurrent malnutrition, sarcopenia, and frailty, compounding their risks. In a study involving adults aged ≥70 with cancer, 33% had upper gastrointestinal and 27% had lung cancer and 83.3% had at least one of the three conditions; 26.7% were identified as having only one condition and 30.0% as having all three conditions simultaneously [28].

## 5. Nutritional Assessment: Screening Tools and Comprehensive Evaluation

### 5.1. Comprehensive Geriatric Assessment

Comprehensive geriatric assessment (CGA) evaluates several domains, including social, demographic information, comorbidities, functional and physical performance, medications, mood and cognitive functions, nutrition, and treatment-associated side effects [29]. The CGA helps clinicians in tailoring cancer treatments to the unique needs of the patient, and it is generally conducted by a geriatrician. Nutritional evaluation in this context is typically a two-step process, composed of a screening phase and a subsequent assessment. Several tools are available to assess patient’s nutritional status and needs.

### 5.2. Geriatric 8

The geriatric 8 (G8) is a short screening tool specifically developed for older cancer patients which comprises eight items covering multiple domains as nutritional status (evaluating food intake, weight loss), mobility, neuropsychological problems, polypharmacy, self-rated health, and age [30,31]. Notably, six of the eight questions are adapted from the Mini Nutritional Assessment (MNA) and focus on nutrition. The G8 is scored 0 to 17 (17 represents best health); a score ≤ 14 is the usual cutoff indicating abnormal result and the need for a subsequent full Comprehensive Geriatric Assessment (CGA). The G8 is recommended by organizations such as the International Society of Geriatric Oncology (SIOG) as an initial screening tool—it is quick (<5 min) and has demonstrated good sensitivity for identifying patients who would benefit from further geriatric assessment. An abnormal G8 has also been associated with worse survival and higher risk of chemotherapy toxicity [32].

### 5.3. Mini-Nutritional Assessment

A validated and useful tool for the identification of older adults at risk of malnutrition is the Mini-Nutritional Assessment (MNA), often used in its short form (MNA-SF), assessing the presence of signs such as reduced food intake, weight loss, mobility, recent acute illness or stress, neuro-psychological problems, and low BMI [33,34]. This tool assigns a total score from 0 to 14, where a score lower than 12 points indicates risk of malnutrition. Diagnosis should be further confirmed by applying etiologic and phenotypic malnutrition criteria as defined by GLIM [6]. Prevalence studies using MNA found a prevalence of malnutrition in approximately 3% in community-dwelling older adults, 22% among hospital inpatients, and nearly 30% in older adults living in nursing homes, long-term care, or rehabilitation/post-acute care settings [35].

### 5.4. Patient-Generated Subjective Global Assessment

The Patient-Generated Subjective Global Assessment (PG-SGA) is an oncology specific nutritional assessment tool which combines a patient-based questionnaire (covering weight history, food intake, symptoms like anorexia or pain, and functional capacity) with a clinician assessment of metabolic demand and physical exam findings. The PG-SGA yields both a global rating (well-nourished A, moderate malnutrition B, severe malnutrition C) and a numeric score that guides the intervention urgency. Scores ≥ 9 indicate a critical need for nutritional intervention (corresponding to severe malnutrition). The PG-SGA is considered a gold-standard assessment in cancer patients and is widely used in clinical trials. However, it is more time-intensive than a screening tool and often requires dietitian involvement. Q. Zhang et al. validated the predictive value of the nutritional assessment tool (Patient-Generated Subjective Global Assessment Short Form, PG-SGA SF) for clinical outcomes in older adults with cancer [36]. Malnutrition, measured by the PG-SGA SF, was found to be a prognostic factor for overall survival in older cancer patients and could improve the prognostic model of TNM [36]. A study by Wang et al. in surgical cancer patients emphasized combining the PG-SGA with objective assessments to accurately diagnose malnutrition in elderly colorectal cancer patients [37]. The PG-SGA incorporates both patient-reported and clinician-evaluated components, making it a practical and effective method for evaluating and managing nutritional issues.

### 5.5. Controlling Nutritional Status Score

The Controlling Nutritional Status (CONUT) Score is an objective screening tool based on laboratory values (serum albumin, total lymphocyte count, and total cholesterol levels) to assess nutritional status [38,39]. Points are assigned for each parameter (e.g., low albumin or lymphopenia yields higher points) and summed into a total score 0–12 (higher scores indicate worse nutritional status). CONUT categories are typically 0–1 normal, 2–4 mild malnutrition, 5–8 moderate, and 9–12 severe. CONUT has been studied as a prognostic indicator in cancers: higher scores correlate with a shorter survival and lower treatment response [40]. This tool is known for its simplicity (requiring just a blood test), but it may be confounded considering that infections or inflammation can also lower albumin and lymphocytes levels. The CONUT is best used in hospitalized or surgical patients where lab values are readily available. A study in 2020 found that this score appears to be an independent prognosticator of performance status (PS) in advanced urothelial cell carcinoma (UCC) patients [41]. Regarding oncological surgery patients, the score was tested as a pre-treatment phase for patients with the colorectal cancer (CRC) and was found that the CONUT score performed as an independent predictor of relapse-free survival (RFS) and OS [38]. It was superior to the Prognostic Nutritional Index (PNI) and to the modified Glasgow Prognostic Score (mGPS) in predicting prognosis of patients who underwent curative surgery for CRC.

### 5.6. Geriatric Nutritional Risk Index

The Geriatric Nutritional Risk Index (GNRI) represents an objective index designed for older adults. It is calculated from weight and albumin (1.489 × albumin [g/L]) + (41.7 × (current weight/ideal weight)) [42]. Typical values range from 60 to 120, with higher scores indicating better nutrition. GNRI can be categorized into risk levels: >98 = no risk; 92–98 low risk; 82–< 92 moderate risk; <82 major risk of nutrition-related morbidity. The GNRI has shown prognostic value in various cancers; it is simple to compute but shows limitations, including the need for an “ideal weight” reference (often defined by a BMI of 22 or the weight at BMI 22). The fact that edema or ascites (common in cancer patients) can confound weight and albumin limits its use. Nonetheless, GNRI is a handy tool in geriatric oncology to flag high-risk patients’ pre-treatment—for example, a patient with GNRI 80 (major risk) should prompt aggressive nutritional support and possibly treatment plan adjustments.

### 5.7. Recommended Approach

A summary of nutritional assessment tools is available in Table 2. Based on geriatric oncology guidelines, CGA should be used to identify vulnerabilities or impairments that are not routinely captured in oncology assessments for all patients over 65 years old with cancer [29,43]. Nutritional screening and assessment, therefore, should not be seen as optional, but rather as a standard-of-care in older patients.

These assessment tools are complementary: a recommended approach is to screen every older cancer patient with a quick tool like G8 (or an alternative such as the MNA-SF or MST—Malnutrition Screening Tool—in non-oncology settings). If the screen is positive (e.g., G8 ≤ 14 or MNA-SF ≤ 11), the clinician should proceed with a more detailed assessment, including a full PG-SGA by a dietitian and a CGA by a geriatrician expert in geriatric oncology, both part of a multidisciplinary team. It is useful to run relevant laboratory blood tests to evaluate specific indices (such as albumin or lymphocytes, which can be used to calculate the CONUT, PNI, and the GNRI and gain additional prognostic insight). Throughout, close follow-up and reassessment are key: weight and performance status are tracked, and the nutrition plan can be adjusted accordingly to implement a proactive approach.

## 6. Nutritional Interventions and Management Strategy

Managing malnutrition and its related syndromes in older cancer patients requires a multimodal and tailored approach due to the heterogeneity of this population in terms of tumor types, disease stages, comorbidities, and individual goals.

### 6.1. Nutrional Counseling

Nutritional counseling represents the first step, giving guidance on meal planning aiming to increase protein and calories intake (such as adding snacks, high-protein beverages, frequent small meals), treating contributory symptoms (providing suggestions for managing taste changes or nausea), and involving caregivers in meal support. An open-label RCT conducted between April 2020 and May 2022 evaluated 80 lung cancer patients, randomizing 43 to dietary counseling and 37 to routine care: the dietary counseling group showed significant benefits, such as smaller decreases in body weight at 3–4 weeks (−0.8% vs. −2.6%, *p* = 0.05) and at 12 weeks too (−1.1% vs. −4.3%, *p* = 0.05). The secondary outcomes such as changes in the BMI, nutrition score, QoL, serum albumin level, lymphocyte count, energy and protein intake, treatment response, PFS, and OS did not differ significantly between the two groups [45].

### 6.2. Oral Nutritional Supplements

Oral nutritional supplements (ONSs), such as high-calorie or high protein drinks, are often prescribed to help patients meet daily requirements when diet alone is not sufficient. ONS could be in different formats (liquid, powder, pudding, and pre-thickened) and types (high in protein, rich in fiber containing). ONS are classified as “high-protein”, when they provide > 20% of energy from protein and “high-energy”, when they provide > 1.5 kcal/mL or gram [46]. Growing evidence supports the efficacy of these measures: a meta-analysis evaluated the efficacy of ONS in cancer patients receiving chemotherapy in recent trials, confirming that ONS could help increase body weight during chemo-treatments and this benefit was especially notable in elderly patients, patients with low baseline body weight, female patients, and non-Asian patients [47].

### 6.3. Management of Symptoms and Therapy-Related Adverse Events 

Another important aspect is represented by symptoms and therapy-related adverse events, which can impact daily dietary intake. The incidence of oral mucositis in patients undergoing chemo-radiotherapy for head and neck tumors ranges from 59.4% to 100%, and tailored interventions should include texture-modified diets, and/or topical analgesics [48,49]. Mucositis and related gastrointestinal toxicities are also common in patients receiving EGFR inhibitors for colorectal and lung cancer: although the risk in monotherapy is low, the relative risk to develop grade ≥ 3 lesions rises significantly when EGFR inhibitors are administrated concomitantly with chemotherapy combination [50,51,52], requiring early recognition, steroids, analgesic and supportive nutritional strategies [53]. Depression or social isolation can lead to anorexia too; appropriate psychosocial support or appetite stimulants might be needed [54].

Diarrhea is frequent with irinotecan-based regimes, with an overall incidence of between 60 and 90% and an incidence of 20–40% for severe diarrhea [55]. The risk is exacerbated in gastrointestinal malignancies such as colorectal and pancreatic cancer, when combination chemotherapies are preferred [56,57,58]. Multidisciplinary management is required for the selection of elderly patients suitable for multidrug regimes and for symptoms management, including antidiarrheal therapy and prophylaxis, hydration, and dietary adjustments [59]. Immune checkpoint inhibitors also affect nutritional status: immune-related colitis impacts 2.4–8.6% of patients and can occur even after treatment interruption [60]. Early diagnosis is crucial, and therapeutic options include steroids, immunosuppressants, or fecal microbiota transplantation [61,62].

Dental issues in older adults such as poor dentition can severely limit intake: referral to a dentist or modification of food consistency can have an immediate impact on nutritional intake [63]. In breast and prostate cancer, the use of antiresorptive agents such as bisphosphonates or denosumab is also associated with osteonecrosis of the jaw, for which a proactive preventive approach is beneficial [64,65,66]. In prostate cancer, androgen deprivation therapy can adversely affect body composition by promoting sarcopenia and weight gain, necessitating long-term management given the prolonged duration of treatment [67].

Polypharmacy is another potential contributor factor, as concomitant medications can cause anorexia or taste changes: simplifying regimens or switching drugs, when possible, may improve appetite and reduce the risk of drug–drug interactions [68,69].

### 6.4. Physical Activity

Malnutrition in older adults often coexists with physical inactivity, and purely nutritional interventions can be less effective if not paired with exercise [70]. Resistance exercise or strength training is recommended whenever feasible, as it helps stimulate muscle protein synthesis and can counteract sarcopenia [71]. Even light resistance bands or tailored physical therapy can help an older patient maintain or improve muscle mass when performed alongside nutritional supplementation. A regular physical exercise regimen during and after cancer treatment has been shown to have both physical and psychological benefits, as decreasing fatigue, improving psychosocial well-being, and, ultimately, improving HrQoL [72,73,74].

The American College of Sports Medicine (ACSM) recommended that patients with cancer and cancer survivors should achieve 150 min per week of moderate intensity aerobic activity [75,76]. The ACSM has also worked on the Exercise Guidelines for Cancer Survivors, Exercise Guidelines for Older Adults, and Guidelines for Exercise Testing and Prescription, providing evidence-based recommendations for designing safe and effective fitness/functional protocols, exercise prescriptions and implementation, and maintenance plans for older adults with cancer and cancer survivors. 

In a recent multicenter phase 3 RCT, patients with colon cancer who completed adjuvant chemotherapy were randomized to a group receiving exercise program and one who received health educational materials only, over a total period of 3 years, for a total of 889 patients [77]. The disease-free survival (DFS) at a median follow-up of 7.9 years was significantly longer in the exercise group than in the other group (HR for disease recurrence, new primary cancer, or death, 0.72; 95% CI 0.55 to 0.94; *p* = 0.02). Moreover, 5-year DFS was 80.3% in the exercise group and 73.9% in the health education one. These results evidence a longer OS in the exercise group (HR 0.63; 95% CI, 0.43 to 0.94). The 8-year overall survival was 90.3% in the exercise group and 83.2% in the health-education group. Notable, musculoskeletal adverse events occurred more in the exercise group (in 18.5%) than in the other group (11.5%).

### 6.5. Pharmacological Agents

In cases of cancer cachexia or when conventional dietary measures fail to counteract weight loss, pharmacologic agents can be considered. Appetite stimulants like megestrol acetate have been used for cancer anorexia: megestrol increases appetite and helps gain weight (mostly fat and water weight) [78]. A Cochrane review in 2013 found inconclusive results regarding the efficacy of megestrol for the treatment of cachexia [79]: another systematic review underscored heterogeneity in the included patients with anorexia/cachexia, related to any pathology (such as cancer, acquired immunodeficiency syndrome), in clinical trials regarding megestrol role in this context [80]. A recent meta-analysis in 2022 found that patients who received high doses of megestrol (>320 mg per day) showed weight loss rather than the expected weight gain, and that megestrol was generally well-tolerated, except for thromboembolic risk, which was more evident with higher doses. Moreover, megestrol did not appear to be effective in providing improvement in cachexia in patients affected by advanced cancer [81].

Another class under study is ghrelin agonists, which seem promising in terms of increasing appetite and lean body mass (LBM) in advanced cancer patients with cachexia. Recent meta-analysis found that anamorelin is associated with an increase in total body weight, LBM, and QOL in patients with cancer-related cachexia, with no increasing overall adverse events [82]. Anamorelin may represent an effective option for cancer cachexia, but further research is required to confirm the efficacy and safety of this drug.

Anti-inflammatory agents, like interleukin-6 (IL-6) inhibitors, have been explored to counteract the inflammatory drive of cachexia: IL-6 targets adipose tissue, skeletal muscle, and gut and liver tissue. Plus, Il-6-sensitive tissues could have their sensitivity increased through a specific mechanism called the “trans-IL6-signaling cascade”, mediated by soluble Il-6 receptors (sIL-6Rs), leading to amplification of its effects, especially in the cachectic patient [83]. In skeletal muscle, chronic IL-6 exposure induces proteasome and autophagy protein degradation pathways, leading to wasting [84,85,86]. IL-6 is also associated with AMPK and NF-κB activation: several mouse cancer models have demonstrated that blocking IL-6 and associated signaling can attenuate cachexia progression [86,87]. IL-6 remains promising as a therapeutic agent in attenuating cachexia within several cancer subtypes, but a better understanding of its direct and indirect effects in cancer patients is mandatory [88].

Some patients may benefit from omega-3 supplementation, which is known to help in gaining weight and in weight maintenance and has been shown to have benefits on muscle [89]. Further studies dedicated to omega-3 fatty acids are desirable to deepen our knowledge regarding their molecular mechanisms and clinical effects on the recovery of muscle homeostasis.

### 6.6. Enteral and Parenteral Nutriotion

Enteral and parenteral nutrition represent suitable options when energetical and nutritional needs of the patients are not meet by oral intake. Enteral nutrition via feeding tube is indicated if the gastrointestinal tract is functional: this is a common scenario in head/neck or esophageal cancers and during chemoradiation to prevent severe weight loss. Parenteral nutrition is generally reserved for those patients where the GI tract is non-functional or obstructed (as in malignant bowel obstruction in ovarian cancer or severe radiation enteritis). In advanced cancer patients, the role of long-term home parenteral nutrition is controversial, but it can be considered on a case-by-case basis. Recent guidelines by ESPEN suggest offering parenteral nutrition to advanced cancer patients only if life expectancy is several months or years and if starvation is the primary threat to life: on the opposite, if expected survival is in a few to several weeks, interventions should be non-invasive, aiming at existential support [90].

## 7. Evidence of Benefits from Nutritional Interventions

Recent studies have started to provide evidence that nutritional support can indeed translate into better outcomes in older or malnourished cancer populations. A secondary analysis of the EFFORT trial (enrolling malnourished medical inpatients, ~20% of whom had cancer) showed that patients receiving individualized nutritional support had fewer serious adverse events [91]. The GAIN study, an RCT, aimed to assess whether geriatric assessment-driven interventions (including involvement of nutritionists for nutritional support) could reduce grade 3 or higher chemotherapy-related toxic effects in older adults with cancer. After enrolling a total of 605 patients starting a new chemotherapy regimen, incidence of grade 3 or higher chemotherapy-related toxic effects was 50.5% in patients receiving a GA-guided intervention vs. 60.6% in patients receiving the standard of care, a total reduction of 10.1% [92].

A network meta-analysis focused on patients with head and neck cancer receiving radiotherapy or chemo-radiotherapy, comparing the effects of different enteral feeding methods such as nasogastric tube (NGT), prophylactic percutaneous endoscopic gastrostomy (pPEG), and reactive percutaneous endoscopic gastrostomy (rPEG). After the selection of 13 studies (a total of 1631 participants), the results showed that both pPEG and NGT were superior to rPEG in the management of the weight loss, and that pPEG was associated with the lowest rate of treatment interruption or nutrition-related illnesses requiring hospital admission. No difference in tube-related complications emerged [93].

Nutritional interventions often result in better patient-reported outcomes: as a 2021 systematic review found, focusing on 18 studies for a total of 2720 subjects included in the analyses, patients receiving nutritional counseling or receiving ONS reported an improved global QoL and physical function. Still, a real specific optimum nutritional intervention was not identified [94].

Preserving muscle mass via a combined nutrition and exercise regimen leads to maintained functional independence, retarding or avoiding becoming wheelchair- or bed-bound. Also, mental health can also improve, as patients with better nutrition often experience less fatigue and depression [95].

Specific dietary regimens such as the Mediterranean Diet (MD) are known for their beneficial effects in cancer incidence: MD is well known to be enriched in antioxidant compounds, associated with promoting anti-inflammatory and metabolic regulatory pathways with several beneficial effects. A recent meta-analysis including 16 studies evaluated the impact of MD in cancer incidence and mortality in older adults, finding that MD could represent a protective factor in cancer prevention in older populations, while no clear effect regarding cancer-related mortality emerged [96].

## 8. Research Gaps and Future Directions

### 8.1. Optimization of Screening Strategy

Despite increased awareness, standardized screening strategies remain lacking. Currently, patients are not always systematically screened for malnutrition: understanding barriers (time, reimbursement, lack of training) and testing pragmatic solutions (such as electronic health record prompts or use of automated screening based on routine data such as BMI trends) represents a priority. Moreover, the adoption of GLIM criteria for malnutrition and a consensus on cachexia criteria remains limited [6].

### 8.2. Interventions Trials

While evidence is growing, to date, intervention studies have had limitations such as small sample sizes or heterogeneity. Currently, there is a need for large RCTs focusing on malnourished or sarcopenic cancer patient groups, especially in the elderly. Studies combining multimodal therapy, including nutrition, physical exercise, and anti-inflammatory drugs, could be particularly promising: a proof of this concept is represented by the Pre-MENAC study, a randomized phase II trial focused on lung and pancreatic cancer patients undergoing chemotherapy who were randomized to standard care alone or to a treatment arm including nutritional interventions, anti-inflammatory drug administration, and a home-based physical exercise regimen (aerobic associated with resistance training). The results showed a mean weight gain of 0.91 kg in the treatment arm and a mean weight loss of 2.12 kg within the control arm at 6 weeks [97].

### 8.3. Patient-Tailored Interventions

Tailoring nutritional interventions to an individual needs is critical, as some patients lose weight primarily from a reduced intake mechanism, for whom high-calorie diets could be helpful, or due to other mechanisms such as hypermetabolism. Precision nutrition includes adjusting protein targets (considering that some older adults might need 1.5 g/kg/day protein to gain muscle, versus the standard 1.0–1.2 g/kg) [46].

In light of this, recently, the European Union proposed the MyPath Nutrition Care Pathway (NCP), a digital platform, aiming to offer an accessible and patient-centered assessment system integrating nutritional, health status, and inflammatory status evaluation in cancer patients; this represents a resourceful solution that can be implemented across several health care settings [98].

## 9. Conclusions

Malnutrition is highly prevalent in older adults with cancer, significantly affecting clinical outcomes, and itoften coexists with sarcopenia, cachexia, and frailty. These four conditions overlap and are frequently underrecognized in clinical practice and their management is often fragmented, despite being potentially actionable from a clinical point of view with timely screening and integrated interventions. In this narrative review, we highlighted the importance of a multidisciplinary approach, emphasizing a two-step screening and assessment process using validated tools. Careful evaluation of each patient’s individual needs should guide targeted interventions, including nutritional counseling and geriatric assessment. We identified three major gaps for future research: (1) optimization of screening strategies, (2) the need for dedicated interventions trials for malnutrition management in elderly patients, and (3) a transition towards a personalized nutrition approach.

In conclusion, proactive identification and integrated management of malnutrition, sarcopenia, cachexia, and frailty should be considered a priority for clinicians, as it can improve quality of life, treatment tolerance, and survival in elderly patients with cancer.

## Figures and Tables

**Table 1 nutrients-17-02928-t001:** Prevalence and clinical impact of nutrition-related conditions in older adults with cancer.

Condition	Definition (Criteria)	Prevalence in Older Adults with Cancer	Clinical Impact (Associated Outcomes)
Malnutrition	Chronic undernutrition resulting in weight loss or low BMI; inadequate calorie/protein intake (GLIM criteria: >5% weight loss or BMI) [6]	From 30 to 60% overall; up to 80% in high-risk groups (as advanced GI cancers) [4]. Approximately 44% are at risk on average.	Increases chemotherapy toxicity and interruptions; postoperative complications (infections, poor wound healing). Decreases treatment response, muscle strength and functional status, and overall survival (higher mortality risk).
Sarcopenia	Age or disease-related loss of skeletal muscle mass and strength (e.g., low muscle mass on CT or BIA plus low grip strength or gait speed per EWGSOP2 criteria) [13,21]	Variable (definition-dependent). Approximately 50% of patients > 70 with solid tumors have sarcopenia; it also occurs in some younger patients (sarcopenic obesity).	Increases chemotherapy dose-limiting toxicities (altered drug distribution); surgical complications (e.g., infections); and longer recovery. Decreases physical function (higher fall risk, loss of independence) and survival (sarcopenia associated with higher mortality in various studies).
Cachexia	Multifactorial wasting syndrome due to disease (e.g., cancer) characterized by involuntary weight loss > 5% (or >2% if BMI	Approximately 50% of all cancer patients develop cachexia; prevalence is 50–80% in advanced or metastatic cancers (especially pancreatic, gastric, lung). Often overlaps with malnutrition and sarcopenia (in the end stage disease, cachexia is common).	Directly causes ~20% of cancer deaths; leads to profound weakness and functional decline; poor tolerance of chemo/radiotherapy (cachectic patients often cannot complete treatment); refractoriness to nutritional support (difficulty reversing weight loss); reduction of QoL (high symptom burden: pain, fatigue, early satiety). Cachexia indicates poor prognosis and often signals limited survival time.
Frailty	Geriatric syndrome of increased vulnerability can be defined by Frailty Phenotype (≥3 of weight loss, exhaustion, weakness, slowness, low activity) or Frailty Index (accumulation of deficits) [16]. Overlaps with malnutrition (weight loss) and sarcopenia (weakness/ slowness)	Approximately 20–40% classified as frail among older cancer patients; another 30–40% prefrail. Prevalence varies by tool and population (frailty higher in advanced disease and in patients > 80).	Increases adverse events from treatment (chemo toxicity, surgical complications, delirium), risk of unplanned hospitalization, and longer hospital stay. Decreases likelihood of completing standard cancer therapies, survival independent of age (frail patients have significantly higher 1-year mortality), and functional independence (higher risk of disability and need for long-term care).

**Table 2 nutrients-17-02928-t002:** Nutritional assessment tools in older cancer patients.

Tool	Type	Content/Parameters	Scoring	Strengths	Limitations
Geriatric 8 (G8) [31]	Screening	8 items (nutrition, mobility, neuropsychology, polypharmacy, self-rated health, age)	0–17 (≤14 = risk)	Quick (<5 min), recommended by SIOG, sensitive	Requires CGA if ≤ 14
Mini Nutritional Assessment - Short Form (MNA-SF) [34]	Screening	Food intake, weight loss, mobility, acute stress, neuropsychological problems, BMI	0–14 (<12 = risk)	Validated in elderly, easy to use	Not cancer-specific; confirmation with GLIM needed
Patient-Generated Subjective Global Assessment (PG-SGA) [36]	Assessment	Weight history, food intake, symptoms (e.g., anorexia, pain), functional capacity + clinical exam	Global rating (A/B/C classes) + Numeric score (≥9 = urgent intervention)	Oncology gold standard, incorporates both patient-reported and clinician-evaluated components incorporates both patient-reported and clinician-evaluated components	Time-intensive, often requires dietitian involvement
Controlling Nutritional Status (CONUT) [38,39,41,44]	Screening	Serum albumin, lymphocyte count, total cholesterol levels	0–12 0–1 = normal, 2–4 = mild 5–8 = moderate 9–12= severe malnutrition	Lab-based, simple	Affected by inflammation/infection
Geriatric Nutritional Risk Index (GNRI) [42]	Prognostic Index	Calculated from weight and albumin Formula: (1.489 × albumin [g/L]) + (41.7 × current/ideal weight)	Typical: 60–120 >98 = no risk; 92–98 = low risk; 82–<92 = moderate risk; <82 major risk of nutrition-related morbidity	Easy to compute	Requires ideal weight, affected by edema/ascites

## Data Availability

Not applicable.

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
