# Peer review of "Nutritional Challenges in Older Cancer Patients: A Narrative Review of Assessment Tools and Management Strategies"

_nutrients, 2025, doi:10.3390/nu17182928_

Round 1
Reviewer 1 Report
Comments and Suggestions for Authors
Dear authors:
Congratulations on your work.
Here are some comments/suggestions:
- The methods described in abstract are not equal to the one described in the methods section;
- Conclusion should be more compreensive, according to the aims formulated in the introduction.
Best regards
Author Response
Dear authors:
Congratulations on your work.
Here are some comments/suggestions:
- The methods described in abstract are not equal to the one described in the methods section;
Response: We thank the Reviewer for pointing this out. We have revised the Abstract to ensure consistency with the "Materials and Methods" section. In detail, we included that this is a narrative review conducted following SANRA guidelines, based on a PubMed search from January 2000 to June 2025, including studies involving older cancer patients (≥65 years) addressing malnutrition, sarcopenia, cachexia, frailty, or nutrition-related interventions.
- Conclusion should be more comprehensive, according to the aims formulated in the introduction.
Response. We appreciate the Reviewer’s suggestion. The Conclusion has been revised to better reflect the scope and aims outlined in the Introduction, including the importance of nutritional interventions and directions for future research.
Reviewer 2 Report
Comments and Suggestions for Authors
Review of the manuscript entitled ‘Nutritional Challenges in Older Cancer Patients: a Narrative Review of Assessment Tools and Management Strategies’ (ID: nutrients-3827828). My comments and suggestions for the manuscript:
- The authors should edit the manuscript in accordance with the journal's editorial guidelines. This applies, among other things, to the abstract, the way references are cited, tables, etc. The guidelines for authors are available on the journal's website.
- In the abstract, the authors should emphasise that this was a narrative review.
- I find that the abstract lacks justification for conducting this review. I ask the authors to supplement it and indicate what gaps in knowledge this review fills.
- In my opinion, the Materials and Methods section needs improvement. The manuscript will be easier to read if the following subsections are separated: Search Strategy and Data Source, Inclusion and Exclusion Criteria, Data Extraction and Quality Assessment, Data Analysis, Synthesis of Results, Study Quality Assessment. The sections should describe the actions taken in detail.
- Was the quality of the research evaluated?
- The study did not include a separate section for results. Scientific texts should include the following sections: Introduction, Materials and Methods, Results, Discussion, and Conclusions.
- The method of selecting articles for the narrative review was not described – preferably using a diagram.
- The studies included in the review were not characterised.
- The quality of the included studies was not assessed using the available scale.
- There is no discussion in this study.
In summary, the manuscript contains too many methodological errors and the presentation of the results is not typical for literature reviews. The authors in this review analyse too much data instead of focusing on 2-3 variables and discussing the review in detail. I suggest that the authors review the literature cited above and rework the manuscript. In my opinion, the current manuscript is not suitable for publication in the journal.
Author Response
Review of the manuscript entitled ‘Nutritional Challenges in Older Cancer Patients: a Narrative Review of Assessment Tools and Management Strategies’ (ID: nutrients-3827828). My comments and suggestions for the manuscript:
1.The authors should edit the manuscript in accordance with the journal's editorial guidelines. This applies, among other things, to the abstract, the way references are cited, tables, etc. The guidelines for authors are available on the journal's website.
Response: We thank the Reviewer for this point. The manuscript has been checked and formatted to ensure compliance with journal’s guidelines. We remain available for further editorial editing.
2.In the abstract, the authors should emphasise that this was a narrative review.
Response: We thank for the point, we have emphasized in the abstract (“Methods” section”), that this was a narrative review.
3.I find that the abstract lacks justification for conducting this review. I ask the authors to supplement it and indicate what gaps in knowledge this review fills.
Response: We thank for this point. In brief, since malnutrition, sarcopenia, cachexia, and frailty can be often underrecognized, and therapeutic management is fragmented, the aim of this review was to synthesize current evidence to favor a multidimensional and personalized nutritional care. We included this statement in the Abstract (Background/Objective) and in the Introduction.
4.In my opinion, the Materials and Methods section needs improvement. The manuscript will be easier to read if the following subsections are separated: Search Strategy and Data Source, Inclusion and Exclusion Criteria, Data Extraction and Quality Assessment, Data Analysis, Synthesis of Results, Study Quality Assessment. The sections should describe the actions taken in detail.
Response: We thank for this point, which helped us to improve the quality of our manuscript. We have modified the Methods section to include a paragraph Search strategy and data source (2.1), Inclusion and Exclusion criteria (2.2) and Data extraction and synthesis of results (2.3). In detail, we specified how the literature search was conducted and how data were extracted. We notice that – being a narrative review – this report does not follow the PRISMA guidelines and therefore not all the points were applicable. According to SANRA guidelines, the description of the literature search includes search terms and inclusion criteria.
5.Was the quality of the research evaluated?
Response: We thank for this point. Since this is not a systematic review and no meta-analysis was performed, and in line with SANRA guidelines, we did not include a formal quality assessment, such as GRADE approach (Grading of Recommendations Assessment, Development and Evaluation). When relevant, the study design was included in the text.
6.The study did not include a separate section for results. Scientific texts should include the following sections: Introduction, Materials and Methods, Results, Discussion, and Conclusions.
Response: We thank for this point. We agree with the reviewer that for original reports an IMRAD format is appropriate. We however notice that narrative reviews often do not follow the IMRAD format rigidly, and our formatting is consistent with other narrative reviews recently published in the journal (e.g. https://doi.org/10.3390/nu17172740).
7.The method of selecting articles for the narrative review was not described – preferably using a diagram.
Response: We thank for point, we acknowledge that a PRISMA flow diagram is required for systematic reviews (as reported in PRISMA). We note that a flow diagram is not mandatory for narrative reviews (and not required by SANRA guidelines). To favor transparency, we specified how the articles assessment was performed in the Methods section.
8.The studies included in the review were not characterised.
Response: We thank the reviewer for this observation. This manuscript is not a systematic review nor meta-analysis and, as such, the aim is not to provide an exhaustive and tabulated characterization of all included studies, but rather to offer an integrative synthesis of the current evidence. To favor transparency, we specified how data were extracted from included studies.
9.The quality of the included studies was not assessed using the available scale.
Response: We thank for this point. As reported in #5, given the non-systematic nature of this analysis, a formal GRADE quality assessment was not performed.
10.There is no discussion in this study.
Response: We thank for this point. As already reported in #6, this review – consistently with other reports (https://doi.org/10.3390/nu17172813)- is not structured following the IMRAD format. Therefore, the appraisal of the included evidence is discussed in the relative subsections, alongside their implications for clinical practice and research.
- In summary, the manuscript contains too many methodological errors and the presentation of the results is not typical for literature reviews. The authors in this review analyse too much data instead of focusing on 2-3 variables and discussing the review in detail. I suggest that the authors review the literature cited above and rework the manuscript. In my opinion, the current manuscript is not suitable for publication in the journal.
Response: We respectfully disagree with these points. Many of the points raised are referred to systematic reviews, whereas it is clearly stated that this was a narrative review. From a methodological point of view, during the revision of the manuscript we checked that all the points required by SANRA guidelines are fulfilled. From a presentation of results point of view, the reporting is consistent with other reports recently published in the journal.
Following reviewers #2 suggestions, changes have been included in the manuscript to improve transparency.
Reviewer 3 Report
Comments and Suggestions for Authors
This review explores the interplay between malnutrition, cancer, and aging, evaluating the prevalence and clinical impact of malnutrition and related conditions in geriatric oncology. This review provides important information. However, the following points should be addressed before publication.
(1) The originality and novelty of this review should be clearly indicated.
(2) Authors should explain more clearly “what is unknown throughout”.
(3) What is the most important impact of this study?
(4) What is the most important point authors want to emphasize in this paper? This paper describes a large amount of information. However, the description seems to be somewhat in a scattered manner.
(5) Citation of the reference papers regarding the five nutritional assessment tools in Table 2 is necessary.
(6) This study investigated the assessment tools and management strategies in older and all types of cancer patients. How about the assessment tools and management strategies in different types of cancer patients? The information of the different cancer types should be incorporated into the text as much as possible.
Author Response
This review explores the interplay between malnutrition, cancer, and aging, evaluating the prevalence and clinical impact of malnutrition and related conditions in geriatric oncology. This review provides important information. However, the following points should be addressed before publication.
1.The originality and novelty of this review should be clearly indicated.
Response: We thank the Reviewer for this observation. We revised Abstract and Introduction to specific that our work synthesizes the overlapping burden of malnutrition, sarcopenia, cachexia, and frailty in older adults with cancer, trying to provide an perspective to a topic that is often fragmented.
(2) Authors should explain more clearly “what is unknown throughout”
Response: We appreciate this comment. In the revised Abstract and Discussion, we now explicitly discuss the knowledge gaps, including: the lack of consensus on a standardized screening approach, limited data on multimodal and personalized interventions in older patients.
(3) What is the most important impact of this study?
Response: We thank for this point. We have clarified in the Conclusion section that the key impact of this review is to provide a reference for assessment and management strategies across malnutrition-related syndromes in older adults with cancer, with the aim of supporting clinicians in the implementation of early screening and favoring multidisciplinary personalized care.
(4) What is the most important point authors want to emphasize in this paper? This paper describes a large amount of information. However, the description seems to be somewhat in a scattered manner.
Response: Thank you for pointing this out. We revised and streamlined the manuscript to improve cohesion and clarity. The key message is that malnutrition, sarcopenia, cachexia, and frailty are highly prevalent but actionable from a clinical point of view, with systematic screening and integrated interventions. We specified this point in the conclusion section.
(5) Citation of the reference papers regarding the five nutritional assessment tools in Table 2 is necessary.
Response: We thank for these points. We have added citations for each of the tools in Table 2.
(6) This study investigated the assessment tools and management strategies in older and all types of cancer patients. How about the assessment tools and management strategies in different types of cancer patients? The information of the different cancer types should be incorporated into the text as much as possible.
Response: We thank the Reviewer for this comment. We acknowledge this is a very important point and therefore we reviewed the manuscript to ensure that tumor-specific details, when applicable, are reported in the text. We also expanded paragraph 6.3 (Symptoms and therapy-related adverse events management) to include more tumor-specific considerations.
Round 2
Reviewer 2 Report
Comments and Suggestions for Authors
Thank you to the authors for reading my review and responding to my suggestions. Of course, I agree that the SANRA review guidelines apply to narrative reviews, which are not systematic reviews. However, I believe that following my suggestions would certainly improve the scientific nature of the study. Including the elements I mentioned in my first review would certainly contribute to a better reception of your manuscript. I find the elements I wrote about in my first review in other narrative reviews that also use the SANRA protocol. However, due to the arguments presented by the authors, I leave the decision on whether to accept the manuscript for publication in the journal to the editor-in-chief. I would be inclined to publish the submitted manuscript, as the corrections already made have increased the scientific value of the work.